# Clinical-Pathological Characteristics of Adenosquamous Esophageal Carcinoma: A Propensity-Score-Matching Study

**DOI:** 10.3390/jpm13030468

**Published:** 2023-03-03

**Authors:** Xinxin Xu, Feng Jiang, Yihan Guo, Hu Chen, Jiayi Qian, Leilei Wu, Dong Xie, Guangxia Chen

**Affiliations:** 1Department of Gastroenterology, The Affiliated Clinical College of Xuzhou Medical University, Xuzhou 221002, China; 2Department of Oncology, Zhongda Hospital, Southeast University, Nanjing 210009, China; 3Department of Scientific Research, Shaanxi Academy of Social Sciences, Xi’an 710061, China; 4Department of Thoracic Surgery, Shanghai Pulmonary Hospital, Tongji University, Shanghai 200092, China; 5Department of Gastroenterology, The Affiliated Xuzhou Municipal Hospital of Xuzhou Medical University, Xuzhou 221002, China

**Keywords:** esophageal adenosquamous carcinoma, propensity score matching, survival, proportion

## Abstract

There are few studies on esophageal adenosquamous carcinoma (ADSC). Our study intended to investigate the clinical and survival features of ADSC. We included esophageal cancer (EC) data from the Surveillance, Epidemiology, and End Results program database to explore clinical and survival traits. Propensity score matching (PSM), the multivariate Cox regression model, and survival curves were used in this study. A total of 137 patients with ADSC were included in our analysis. The proportion of ADSC within the EC cohort declined from 2004 to 2018. Besides, results indicated no significant difference in survival between ADSC and SCC groups (PSM-adjusted HR = 1.249, *P* = 0.127). However, the survival rate of the ADSC group was significantly worse than that of the ADC group (PSM-adjusted HR = 1.497, *P* = 0.007). For the ADSC group, combined treatment with surgery had a higher survival rate than other treatment methods (all *P* < 0.001). Surgical resection, radiotherapy, and chemotherapy were independent protective prognostic factors (all *P* < 0.05). The proportion of ADSC has been declining from 2004 to 2018. The prognosis of ADSC is not significantly different from that of SCC but is worse than that of ADC. Surgery, radiotherapy, and chemotherapy could improve the prognosis of patients. Comprehensive treatment with surgery as the main treatment is more beneficial for some patients.

## 1. Introduction

The latest global cancer statistics indicated that esophageal cancer (EC) is one of the most common primary gastrointestinal malignancies throughout the world [1]. It is reported that EC was the eighth-most common malignancy (new EC cases, N = 604,000) and the sixth-most common cause of cancer-related mortality (new EC deaths, N = 544,000) worldwide in 2020 [2]. Common pathologic types of EC contain esophageal squamous cell carcinoma (SCC) and adenocarcinoma (ADC). In addition, the incidence of EC varies geographically among different subtypes [3]. SCC incidence is highest in East Asia, while ADC incidence is highest in Northern Europe. Although SCC remains the most common type of EC worldwide, the incidence of ADC exceeds that of SCC in some high-income countries [4]. As early clinical symptoms of EC are not obvious, most patients are in the middle- and late-stage disease when they visit the hospital. The traditional treatments for EC are surgical resection, radiotherapy, chemotherapy, and molecular targeted therapy. In recent years, promising treatments for EC are emerging, but the prognosis of patients remains unsatisfactory and the 5-year overall survival (OS) rate still ranges from 15 to 25% [5]. Therefore, it is essential to evaluate the prognoses of EC and identify prognostic features in patients.

Primary adenosquamous carcinoma (ADSC), characterized by low incidence, high invasiveness, and poor prognosis, is an extremely rare histological type of EC [6]. Although the biological characteristic of ADSC is similar to that of ADC and SCC, it is not a simple fusion of these two types. There is no consensus in the current literature regarding the diagnostic criteria for ADSC in the digestive system. The 2010 edition of the *World Health Organization Classification of Gastrointestinal Neoplasms* did not define the proportion of ADC and SCC in the diagnosis for ADSC [7]. According to the criteria established by the Japanese Society of Esophagus, the diagnosis criteria of esophageal ADSC are two common histological types (ADC and SCC) accounting for more than 20% each [8]. This rare histological type accounts for about 0.9% of EC [9]. Because of the low incidence rate of ADSC, previous studies have mainly focused on ADC and SCC [10,11,12]. Currently, there are not many studies on the clinicopathological characteristics and survival analysis of ADSC in the world, so in this study, we decided to use the method of propensity score matching (PSM) to contrast the clinical characteristics and prognosis of ADSC and other two common tumor types from 2004 to 2018 using the Surveillance, Epidemiology, and End Results (SEER) program database, aiming to provide some reference value for clinicians’ judgment on the improvement of the diagnosis and treatment levels and prognosis of ADSC.

## 2. Materials and Methods

### 2.1. Patient Selection

The SEER database is the cancer epidemiology and official source of cancer statistics. The SEER database includes cancer incidence and survival population data, relying on cancer registries in 19 geographic areas of the United States, covering morbidity, mortality, and prevalence information for approximately 35% of the US population [13]. Data on nine types of cancer cases are included in the SEER database, and cancer incidence is compared to the proportion of the population living. Through the analysis of these data, we can assimilate the cancer burden and its development trend and provide a basis for the formulation of prevention and control policies.

The Ethics Committee of Xuzhou Medical University Affiliated Municipal Hospital ratified the study, claiming that it dispensed with an ethical review because existing data on patient identity information were hidden. Informed consent was not required. We studied the patients in the SEER database who were definitively diagnosed with EC from 2004 to 2018. These patients were over 19 years old. In total, 44,948 patients with EC were involved after initial case selection. For further analysis of survival, our inclusion criteria were as follows: diagnosed as ADC, SCC, or ADSC tumors. We excluded patients who were diagnosed between 2016 and 2018, died within 1 month of diagnosis, or had less than 5 years of follow-up for inventory status. As SEER database information is currently only updated until 2020, cases from 2016 to 2018 were excluded because the 5-year survival rate was not available. The complete selection criteria are in Figure 1. Basic information gathered from the SEER database included sex, race/ethnicity, marital status, age at diagnosis, diagnosis year, and survival time. Tumor characteristics included the primary location (upper, middle, and lower thoracic esophagus), differentiation degree, TNM classification, histological subtype, and therapeutic method (e.g., surgery, radiotherapy, and chemotherapy). Overall survival (OS) and cancer-specific survival (CSS) were considered as the primary end points of this study.

### 2.2. Propensity Score Matching

In this paper, gender, age, marital status, tumor site, TNM stage, and other pathological characteristics were used as matching variables. The propensity matching score was achieved using SPSS and SPSSAU, and the correction threshold was set as 0.1. The difference in baseline characteristics between the two groups was matched in a 1:1 ratio.

### 2.3. Statistical Analysis

All statistical analysis was performed using SPSS Statistics 25.0 software (IBM SPSS, Inc., Armonk, IL, USA) and GraphPad Prism 8 (Version 8.4.3). The chi-square test was performed to compare the differences in clinical baseline characteristics. The trend in the proportion of ADSC in the entire cohort was plotted. Risk ratios (RRs), and hazard ratios (HRs) and 95% confidence intervals (CIs) were calculated using multivariable logistic regression analysis and Cox regression analysis, respectively (the regression method was Enter selection). A multivariate Cox regression model was established, and survival curves were drawn to determine the survival in EC of different pathological types. Multivariable Cox regression analysis was conducted according to the results of the univariable analysis to determine independent prognosis factors. To prevent the statistical results from being affected by missing data, we also performed Cox regression analysis and plotted survival curves using the original data of 44,948 patients with EC to confirm the reliability of the results. Two-sided *P* < 0.05 was considered statistically significant.

## 3. Results

### 3.1. Patient Characteristics

Our study initially identified 44,948 cases diagnosed with EC from 2004 to 2018. After screening, we finally included 17,785 EC cases for survival analysis, including 137 ADSC cases, of which 100 patients with ADSC died in the cohort at the end of follow-up. The number of male patients with ADSC (N = 106) was more than that of females (N = 31), and the age distribution was homogenized in this cohort. We analyzed the anatomical location of ADSC, and the data showed that ADSC mainly occurred in the lower thoracic esophagus (N = 92, 67.2%), followed by the middle thoracic esophagus (N = 34, 24.8%). In accordance with the 8th edition of the *American Joint Committee on Cancer* staging system for SCC, 38 (27.7%) cases of cancers invaded the musculature propria, the musculature, or the submucosa (T1); 15 cases (10.9%) invaded the musculature propria (T2); 62 cases (45.3%) involved the tunica adventitia (T3); and 22 cases (16.1%) invaded local structures (T4). For the N stage, 81 (59.1%) patients had lymph node metastasis (N+). The detailed characteristic information about the patients is presented in Table 1. Finally, 17,785 patients met our selection criteria and were further analyzed using PSM. There were significant differences in sex, age, race, marital status, location, TNM stage, surgery, and grade before matching (Table 1). We adopted the PSM method to balance the differences in baseline characteristics between the groups. Finally, 411 patients were selected according to the ratio of 1:1, including 137 patients in the ADSC group, 137 patients in the SCC group, and 137 patients in the ADC group. A comparison of the baseline characteristics between the matched groups showed that the differences in baseline characteristics between the groups significantly reduced, as shown in Table 2.

### 3.2. Proportion of ADSC

The proportion of ADSC in the entire cohort was 0.7 per 100 patients and decreased from 2004 to 2018 (Figure 2; 0.834 per 100 patients (95% CI 0.005–0.012) in 2004; 0.548 per 100 patients (95% CI 0.003–0.008) in 2018). Among the 44,948 cases in the EC cohort, the largest proportion of ADC showed a continuous increase year by year, with a slow decrease in the proportion of SCC (Figure 3). Multivariable logistic regression indicated that the proportion of ADSC presented a declined trend over the year (Table 3; adjusted RR = 0.972, 95% CI 0.947–0.998, *P* = 0.037). In addition, linear regression analysis led to the same results (R = 0.578, *P* < 0.05). In addition, the high incidence rate among Caucasians was significant (adjusted RR = 1.461, 95% CI 1.045–2.044, *P* = 0.027).

### 3.3. Survival Analysis of Histology Classification

Our results showed that the survival rate of EC was at a low level. We found no significant difference in survival between the ADSC group and the more common histological classification of EC in the results of the multivariable analysis before PSM. Patients with SCC (unadjusted HR = 1.081, *P* < 0.05) had a worse prognosis than patients with ADC. The unadjusted survival curves were consistent with these results (Figure 4A). After PSM, data analysis showed that the median survival time of ADC, SCC, and ADSC was 15 months (95% CI 11.0–20.0 months), 12 months (95% CI 10.0–15.0 months), and 11 months (5% CI 9.0–13.0 months), respectively. Besides, further analysis found that the survival rate of patients with ADSC (PSM-adjusted HR = 1.497, *P* = 0.007) was lower than that of patients with ADC (Figure 4B). We found that the SCC group had a higher survival rate than the ADSC group, but the difference was not statistically significant (PSM-adjusted HR = 1.249, *P* = 0.127; Figure 4C). The PSM-adjusted survival curve was consistent with these results. We performed survival analysis again using the pre-deletion data and obtained results that were generally consistent with our selection of patients by condition (see Appendix A).

### 3.4. Survival Analysis of Different Treatment Groups in ADSC

According to the different treatment methods, patients with ADSC were divided into four groups: patients without surgery, radiotherapy, and chemotherapy were categorized into the untreated group (N = 9), patients who only underwent surgery without radiotherapy or chemotherapy were categorized into the surgery-only group (N = 15), patients without surgery but radiotherapy or chemotherapy were categorized into the chemoradiotherapy group (N = 79), and patients who underwent surgery, radiotherapy, and chemotherapy were categorized into the combined treatment group (N = 33). Survival analysis was performed on these four groups of patients (Figure 5). Cox survival curve analysis showed that the survival rate of patients in the combined treatment group was significantly higher than that in the other three groups (*P* < 0.001). The survival rate of patients in the surgery-only group was higher than that in chemoradiotherapy group. These results further confirmed that surgery combined with chemoradiotherapy has a higher clinical application value than chemoradiotherapy alone in patients with ADSC.

### 3.5. Prognostic Analysis for ADSC

About one-third of the 137 patients with ADSC underwent surgery. The number of patients who received radiation therapy was 95 (69.3%), the same as those who received chemotherapy. The relationship between clinicopathological features and prognosis is presented in Table 4. Among them, surgical resection, radiotherapy, and chemotherapy could significantly improve the outcomes of patients (adjusted HR = 0.326, 95% CI 0.168–0.631, *P* = 0.001; adjusted HR = 0.532, 95% CI 0.299–0.949, *P* = 0.032; and adjusted HR = 0.409, 95% CI 0.215–0.777, *P* = 0.006, respectively). By adjusting for confounders, we identified surgical resection, radiotherapy, and chemotherapy as independent prognostic factors (Table 4). In addition, there were no significant differences in mortality by patients’ sex, age, marital status, tumor differentiation, location of the tumor, and TNM classification in both univariable and multivariable analyses (all *P* > 0.05). The results of the Cox survival analysis of the 380 patients with ADSC before censoring were consistent with those after screening (see Appendix A).

## 4. Discussion

ADSC is a relatively rare tumor. Besides, there is a lack of research to inform treatment recommendations and prognostic assessments for patients with ADSC. We used a large database to investigate the clinicopathological characteristics and prognostic factors of patients with ADSC. In addition, the patient outcomes for different treatment modalities are of great interest to clinicians in developing treatment plans and assessing prognosis. In this study, we used the data of 44,948 patients with EC to calculate the proportion of ADSC cases. The proportion of ADSC was 0.7% in our study, and the proportion of ADSC was on the decline between 2004 and 2018. Besides, 137 patients with ADSC were included in baseline characteristics analysis and further Cox regression analyses. According to the results, the survival rates in ADSC were similar to those in SCC but significantly lower than those in ADC. Gender, age, grade, and marital status were not independent protective prognostic factors for ADSC. By adjusting for confounders, surgical resection, radiotherapy, and chemotherapy were identified as independent prognostic factors. In addition, combined treatment with surgery was more effective than other treatment options in improving the survival rate. Therefore, we believe that comprehensive treatment based on surgical resection is the optimal treatment for some patients with ADSC.

The prognosis of esophageal malignant tumors is poor. Because their symptoms are hidden, most patients are already in the middle and late stages of clinical treatment. In this study, the disease of most patients was in the medium or low differentiation level, and more than half of them had metastasis of the lymph nodes. For patients with this disease staging, surgical treatment is often not recommended, which leads to the 5-year survival rate of the patients being less than 20% [14]. At present, ESCC mainly occurs in two-thirds of the stratified squamous epithelium of the esophagus. Histopathology can observe squamous differentiation of the esophageal epithelium; damage to basal and underlying structures; and matrix reactions, vascular lymphatic vessels, and peripheral nerves [15]. EC is common in men. A previous study found that sex hormone levels may be a factor affecting the gender difference in EC incidence [16]. In addition, EC is directly related to alcohol and tobacco consumption. Long-term tobacco and alcohol stimulation changes the tissue microenvironment, which can greatly increase the risk of cloning and evolution of cancer cells [17]. ADC usually occurs in the lower third of the esophageal mucosa, mainly originating from the Barrett mucosa. Since the 1970s, the incidence of ADC in Europe and the United States has gradually increased. One of the reasons may be the increase in the proportion of people with obesity, which increases the prevalence of gastroesophageal reflux. The lower esophageal mucosa is in a low-pH environment for a long time, and the risk of ADC also increases [2]. ADSC is more invasive than epidermal cancer, with a spread of more than 75% in local lymph nodes, distant metastasis of more than 25%, and a 5-year survival rate of 15–25% [15]. In fact, regardless of histology, the prognosis level of esophageal cancer is unsatisfactory. Only about 20% of patients can survive for 3 years or more after diagnosis [18]. Therefore, primary prevention and early diagnosis of EC require successful strategies. Primary prevention of EC mainly involves lifestyle changes, such as avoiding smoking and drinking alcohol; secondary preventions, such as endoscopy monitoring programs and chemical prophylaxis, are equally important.

ADSC is a special mixed cell type of EC and a comparatively scarce pathological type of EC, which means that the cancer tissue incorporates both SCC and ADC components. Due to the extremely low incidence of ADSC, most of the current studies on ADSC are isolated case reports or clinicopathological series reports [19,20,21,22]. However, the specific incidence and prognostic characteristics of ADSC in different countries or regions are not well described. According to the current literature, the proportion of ADSC is between 0.37 and 1% [23]. This study showed that the number of ADSC cases account for about 0.7% (300/44,948) of primary EC, which is basically consistent with literature reports. Furthermore, the proportion of ADSC in the entire EC alignment decreased from 2004 to 2018. The promotion of computed tomography and upper gastrointestinal endoscopy may have contributed to the decline in the overall trend [24,25,26,27].

However, the accuracy of upper gastrointestinal endoscopic biopsy in diagnosing ADSC is low. Preoperative endoscopic biopsy just diagnosed 1 (4.2%) case of ADSC in the study by Sun et al. [28]. Most patients were diagnosed with SCC on preoperative biopsy [20,29]. This might be because the SCC component was located in the mucosa, while the ADC component was mainly located in the deeper region of the tumor or in metastatic lymph nodes [23]. Another important reason was that almost all ADSCs were covered by normal epithelial cells or intraepithelial neoplasia [30]. A combination of magnifying endoscopy with narrow-band imaging (ME-NBI), endoscopic ultrasonography (EUS), and deeper biopsy potentially improves preoperative diagnostic accuracy [31].

At present, due to a lack of cases, there is no large sample of prospective randomized control results to clarify the biological characteristics and prognosis of ADSC, and there is also disagreement about the survival and prognosis in ADSC. Existing studies are controversial in comparing the prognosis in ADSC, ADC, and SCC. ADSC has been proven to be more aggressive than ADC and SCC alone, with a high rate of lymph node metastasis and inferior prognosis [21,32]. Most of the previous studies have found that the OS of patient with ADSC is worse than that of patient with ADC or SCC [19,33]. Data from Evans et al. showed that the 2- and 5-year OS rates in ADSC are 23.8 and 12.8%, respectively, which is lower in ADC and SCC [9]. Our results showed that ADSC survival is similar to SCC survival but inferior to ADC survival. We hypothesized that although ADSC confuses two different tumor components, the prognosis is not a general combination of the two. Coincidentally, Yendamuri et al. and Chen et al. reported that the prognosis of ADSC is consistent with that of SCC [34,35]. In addition, Yachida et al. found that ADSC had better prognosis than SCC in their series of 18 patients with this disease [36]. This paradoxical result might be explained by the fact that the ADSC tumors included were early tumors with a low T status. Another possibility is that due to the small number of ADSC cases, there was selection bias of clinical factors, and sample size analysis needs to be further expanded to improve the accuracy.

Surgery, radiotherapy, and chemotherapy are the main treatment approaches for patients with EC [37,38], which were recognized as independent protective predictors of prognosis in our results. The prognosis of patients undergoing surgery and chemotherapy were better in both single-factor analysis and multi-factor analysis. Patients undergoing surgical resection have a superior prognosis, which is related to the clinical stage of ADSC, and in general, patients who get an opportunity for surgery tend to have earlier stages of the disease. In this study, 137 patients with ADSC were analyzed, and it was found that the prognosis was correlated with the selection of treatment methods and the difference was statistically significant. The survival time of patients treated with surgery combined with radiotherapy or chemotherapy was generally longer than that of patients treated with surgery alone. This may be because ADSC easily invades the surrounding tissues and organs, so surgical resection alone often cannot completely remove the cancer foci. Preoperative or postoperative chemoradiotherapy may help to remove the residual lesions, thus improving the survival time of patients. Therefore, it is believed that the preferred treatment of ADSC should be surgery assisted with local radiotherapy or chemotherapy, which is also a relatively recognized treatment method at present [39,40,41]. It is important to note that the cloning dynamics of cancer are altered by the introduction of drugs or radiation therapy, with consequent mass cell death, providing selective pressure on the proliferation of mutated cells. However, many chemotherapy drugs are genotoxic, and surviving cancer cells may develop genetic mutations that increase their fitness and malignancy potential. In addition, in recent years, drugs targeting the immune checkpoint mechanism are a new and widely used and effective means [42,43]. A large number of studies have shown that immunotherapy is effective for EC, a tumor with a high mutation burden, and has now become another effective treatment method for middle and advanced esophageal malignancies after surgical resection, radiotherapy, and chemotherapy.

Recent studies have reported that lymph node metastasis is the exclusive independent prognostic factor for ADSC [29,44]. In 2016, Qian et al. believed that after correcting the interference of other mixed factors, a higher combined stage, a frequent lymph node metastasis rate, and higher distant metastasis are associated with poor prognosis of ADSC [33]. Contrary to the findings of this previous study, our series did not find the tumor location, TNM stage of the tumor, and tumor grade affecting the prognosis of ADSC. The reason may be that some data were limited by the availability of the database. On average, 5.1% of cancers were classified as overlapping or unspecified locations and 15.3% of the patients were not clearly graded. Another reason is that the sample size included was relatively small, and the baseline level of included cases was uneven.

Our research still leaves something to be desired. First, we could not obtain some crucial information, such as distant metastases, the serous membrane invasion area, and the tumor infiltration depth, because we could not obtain exhaustive data of esophagoscopy and computerized tomography examination from the SEER database. Therefore, we did not conduct an in-depth classification and evaluation of ADSC. Second, this study lacked specific diagnosis and treatment information. We only compared the prognosis after surgical resection, radiotherapy, and chemotherapy, and the exact combination and sequence of treatments are unclear. Finally, since this study was retrospective, expanding the sample size analysis is required to further explore and verify the incidence, clinicopathological features, diagnosis, and prognostic factors of patients with ADSC.

## 5. Conclusions

ADSC is a special type of tumor different from SCC and ADC, and its prognosis is similar to that in SCC but significantly lower than that in ADC. Surgery, radiotherapy, and chemotherapy can improve the prognosis of patients. Comprehensive treatment with surgery as the main treatment is more beneficial for some patients. Further large-scale or multi-center randomized prospective trials are required to explore the biological behavior of esophageal ADSC, improve the accuracy of early diagnosis, and set more effective diagnostic guidelines and therapeutic strategies.

## Figures and Tables

**Figure 1 jpm-13-00468-f001:**
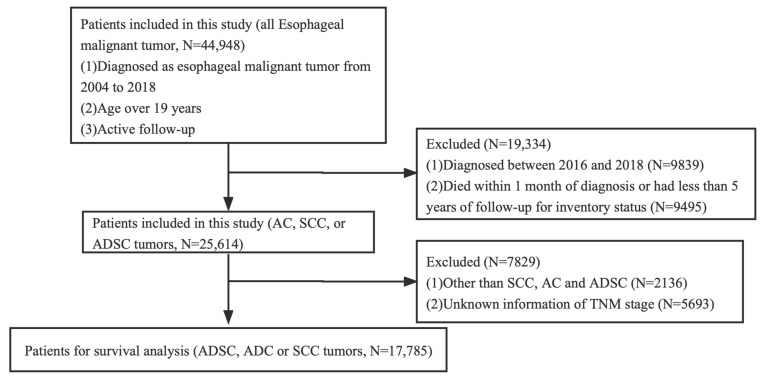
The flowchart of this study.

**Figure 2 jpm-13-00468-f002:**
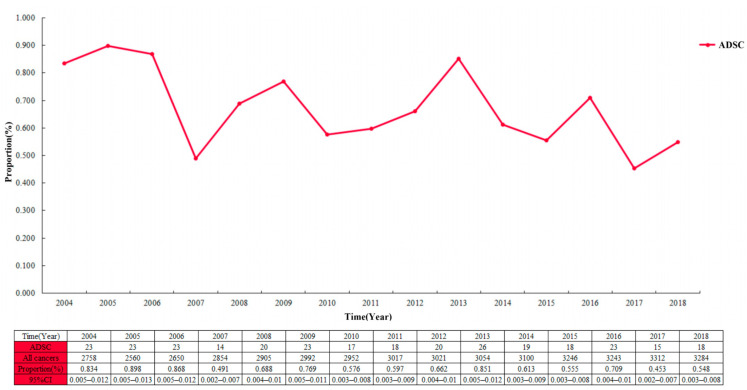
The proportion of esophageal adenosquamous carcinoma over time in the 44,948 patients with esophageal cancer.

**Figure 3 jpm-13-00468-f003:**
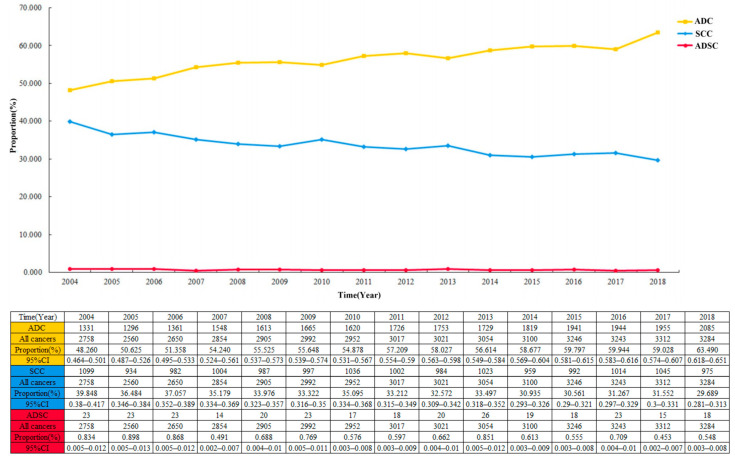
The crude incidence rate of three different pathological types of esophageal cancer over time in 44,948 patients.

**Figure 4 jpm-13-00468-f004:**
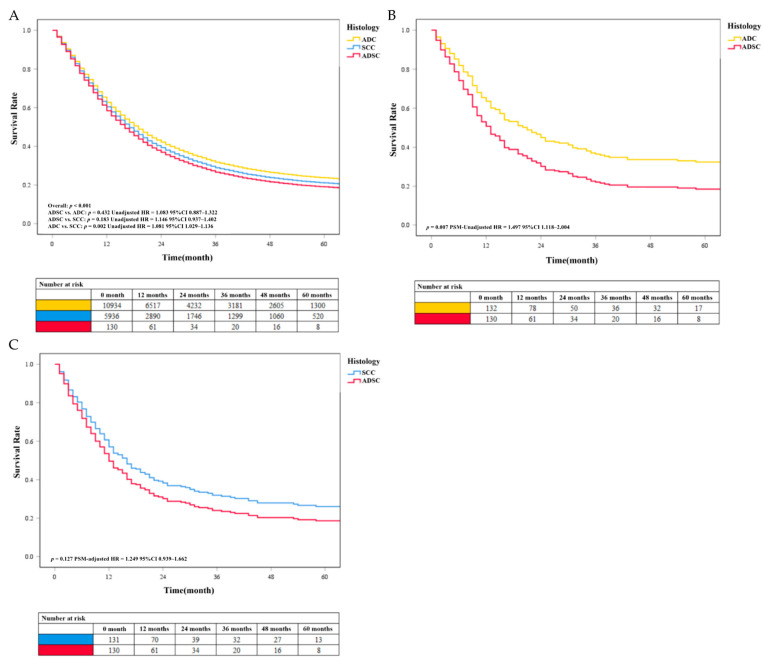
(**A**) The unadjusted survival curves of different histology classifications before PSM. The COX survival curves of EC with different histology classifications after PSM. (**B**) Survival comparison between the ADC and ADSC groups. (**C**) Survival comparison between the SCC and ADSC groups.

**Figure 5 jpm-13-00468-f005:**
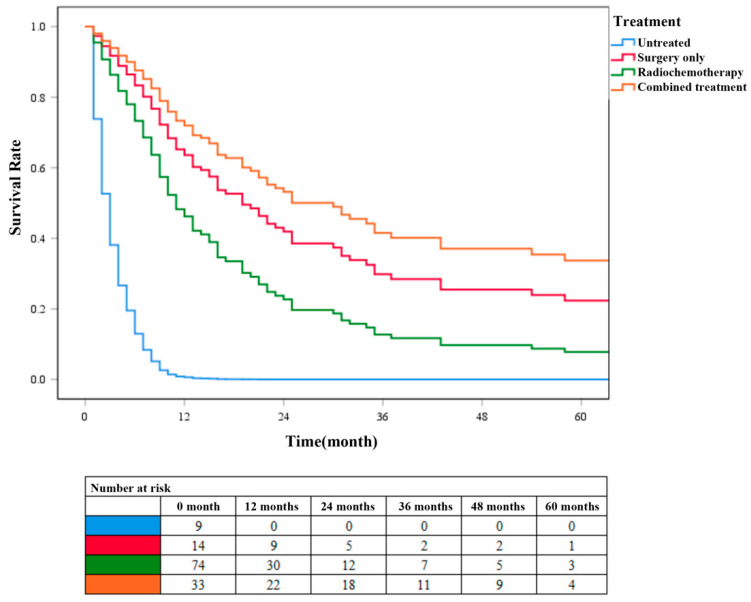
COX survival curves for different treatment groups of patients with ADSC.

**Table 1 jpm-13-00468-t001:** Baseline characteristics of three different pathological types of esophageal cancer in the SEER database (N (%)).

	ADSC	SCC	ADC	χ^2^	*P*-Value
	**(N = 137)**	**(N = 6220)**	**(N = 11,428)**		
Sex				1157.654	<0.001 ^ab^
Female	31 (22.6%)	2177 (35.0%)	1509 (13.2%)		
Male	106 (77.4%)	4043 (65.0%)	9919 (86.8%)		
Age				6.816	0.033
<65 years	57 (41.6%)	2646 (42.5%)	5090 (44.5%)		
>64 years	80 (58.4%)	3574 (57.5%)	6338 (55.5%)		
Race				3493.035	<0.001 ^ab^
White patients	119 (86.9%)	3692 (59.4%)	10,838 (94.8%)		
Other patients	18 (13.1%)	2528 (40.6%)	590 (5.2%)		
Marital status				543.109	<0.001 ^b^
Unmarried	62 (45.3%)	3150 (50.6%)	3805 (33.3%)		
Married	72 (52.6%)	2795 (44.9%)	7154 (62.6%)		
Unknown	3 (2.2%)	275 (4.4%)	469 (4.1%)		
Location				6441.786	<0.001 ^ab^
Upper thoracic esophagus	4 (2.9%)	1002 (16.1%)	117 (1.0%)		
Middle thoracic esophagus	34 (24.8%)	2634 (42.3%)	733 (6.4%)		
Lower thoracic esophagus	92 (67.2%)	2060 (33.1%)	10,342 (90.5%)		
Unknown	7 (5.1%)	524 (8.4%)	236 (2.1%)		
Radiotherapy				149.737	<0.001
No	41 (29.9%)	1742 (28.0%)	4239 (37.1%)		
Yes	95 (69.3%)	4415 (71.0%)	7095 (62.1%)		
Unknown	1 (0.7%)	63 (1.0%)	94 (0.8%)		
Chemotherapy				5.637	0.060
No	42 (30.7%)	2056 (33.1%)	3580 (31.3%)		
Yes	95 (69.3%)	4164 (66.9%)	7848 (68.7%)		
Surgery				771.160	<0.001 ^a^
None	89 (65.0%)	4887 (78.6%)	6637 (58.1%)		
Local treatment	1 (0.7%)	109 (1.8%)	607 (5.3%)		
Surgical resection	46 (33.6%)	1190 (19.1%)	4089 (35.8%)		
Unknown	1 (0.7%)	34 (0.5%)	95 (0.8%)		
Grade				164.213	<0.001 ^ab^
I	0 (0.0%)	318 (5.1%)	610 (5.3%)		
II	20 (14.6%)	2627 (42.2%)	4031 (35.3%)		
III–IV	96 (70.1%)	2210 (35.5%)	4906 (42.9%)		
Unknown	21 (15.3%)	1065 (17.1%)	1881 (16.5%)		
T				109.641	<0.001
T1	38 (27.7%)	2087 (33.6%)	3955 (34.6%)		
T2	15 (10.9%)	780 (12.5%)	1471 (12.9%)		
T3	62 (45.3%)	2270 (36.5%)	4638 (40.6%)		
T4	22 (16.1%)	1083 (17.4%)	1364 (11.9%)		
N				18.875	0.016
N0	56 (40.9%)	3023 (48.6%)	5258 (46.0%)		
N+	81 (59.1%)	3197 (51.4%)	6170 (54.0%)		
M				71.458	<0.001
M0	98 (71.5%)	5048 (81.2%)	8653 (75.7%)		
M1	39 (28.5%)	1172 (18.8%)	2775 (24.3%)		

Note: Compared with the SCC group, ADSC group ^a^ *P* < 0.05; compared with the ADC group, ADSC group ^b^ *P* < 0.05.

**Table 2 jpm-13-00468-t002:** Demographic and clinical characteristics of the three groups after PSM (N (%)).

	ADSC (N = 137)	SCC (N = 137)	ADC (N = 137)	χ^2^	*P*
Sex				2.450	0.294
Female	31 (22.6%)	33 (24.1%)	23 (16.8%)		
Male	106 (77.4%)	104 (75.9%)	114 (83.2%)		
Age				2.187	0.335
<65 years	57 (41.6%)	68 (49.6%)	58 (42.3%)		
>64 years	80 (58.4%)	69 (50.4%)	79 (57.7%)		
Surgery				4.541	0.604
None	89 (65%)	94 (68.6%)	85 (62%)		
Local treatment	1 (0.7%)	1 (0.7%)	0 (0%)		
Surgical resection	46 (33.6%)	42 (30.7%)	52 (38%)		
Unknown	1 (0.7%)	0 (0%)	0 (0%)		
Radiotherapy				4.036	0.401
No	41 (29.9%)	32 (23.4%)	41 (29.9%)		
Yes	95 (69.3%)	105 (76.6%)	96 (70.1%)		
Unknown	1 (0.7%)	0 (0%)	0 (0%)		
Chemotherapy				3.602	0.165
No	42 (30.7%)	32 (23.4%)	29 (21.2%)		
Yes	95 (69.3%)	105 (76.6%)	108 (78.8%)		
Marital status				5.799	0.215
Unmarried	62 (45.3%)	64 (46.7%)	52 (38%)		
Married	72 (52.6%)	71 (51.8%)	85 (62%)		
Unknown	3 (2.2%)	2 (1.5%)	0 (0%)		
Grade				10.387	0.109
I	0 (0%)	0 (0%)	4 (2.9%)		
II	20 (14.6%)	28 (20.4%)	28 (20.4%)		
III–IV	96 (70.1%)	90 (65.7%)	87 (63.5%)		
Unknown	21 (15.3%)	19 (13.9%)	18 (13.1%)		
Location				23.427	0.001 ^b^
Upper thoracic esophagus	4 (2.9%)	8 (5.8%)	1 (0.7%)		
Middle thoracic esophagus	34 (24.8%)	39 (28.5%)	17 (12.4%)		
Lower thoracic esophagus	92 (67.2%)	84 (61.3%)	117 (85.4%)		
unknown	7 (5.1%)	6 (4.4%)	2 (1.5%)		
T				6.892	0.331
T1	38 (27.7%)	46 (33.6%)	41 (29.9%)		
T2	15 (10.9%)	16 (11.7%)	10 (7.3%)		
T3	62 (45.3%)	54 (39.4%)	72 (52.6%)		
T4	22 (16.1%)	21 (15.3%)	14 (10.2%)		
N				3.683	0.451
N0	56 (40.9%)	59 (43.1%)	60 (43.8%)		
N1	3 (2.2%)	0 (0%)	1 (0.7%)		
N2	0 (0%)	0 (0%)	0 (0%)		
N3	78 (56.9%)	78 (56.9%)	76 (55.5%)		
N+	0 (0%)	0 (0%)	0 (0%)		
M				3.992	0.136
M0	98 (71.5%)	110 (80.3%)	97 (70.8%)		
M1	39 (28.5%)	27 (19.7%)	40 (29.2%)		

Note: Compared with the ADC group, ADSC group ^b^ *P* < 0.05.

**Table 3 jpm-13-00468-t003:** The results of multivariable logistic regression analyses.

	Multivariable Analysis		
	RR	95% CI	*P*-Value
Sex (male = 1 vs. female = 0)	1.115	0.838–1.484	0.455
Year of diagnosis (continuous)	0.972	0.947–0.998	0.037
Age (<65 years = 0 vs. ≥65 years = 1)	0.910	0.722–1.147	0.424
Race (Caucasians = 1 vs. others = 0)	1.461	1.045–2.044	0.027

Note: The results of multivariable analysis were adjusted for other confounding factors, such as sex, age, and race/ethnicity. RR: risk ratio; CI: confidence interval. The logistic regression method was Enter selection.

**Table 4 jpm-13-00468-t004:** Univariable and multivariable Cox proportional hazard regression analyses for mortality in 137 patients with ADSC.

Variables		Univariable Analysis		Multivariable Analysis		
	HR	95% CI	*P*	HR	95% CI	*P*
Sex						
Female	1	Reference		1	Reference	
Male	0.952	0.600–1.509	0.833	1.010	0.595–1.717	0.969
Age						
<65 years	1	Reference		1	Reference	
>64 years	0.951	0.641–1.409	0.801	0.742	0.477–1.153	0.184
Surgery						
None	1	Reference		1	Reference	
Local treatment	1.428	0.197–10.361	0.725	3.930	0.352–43.871	0.266
Surgical resection	0.419	0.269–0.653	0	0.326	0.168–0.631	0.001
Unknown	0.596	0.082–4.315	0.608	0.127	0.006–2.738	0.188
Radiotherapy						
No	1	Reference		1	Reference	
Yes	0.656	0.430–1.002	0.051	0.532	0.299–0.949	0.032
Unknown	7.593	0.977–59.007	0.053	3.703	0.407–33.716	0.245
Chemotherapy						
No	1	Reference		1	Reference	
Yes	0.627	0.408-0.965	0.034	0.409	0.215-0.777	0.006
Marital status						
Unmarried	1	Reference		1	Reference	
Married	0.709	0.476–1.057	0.092	1.016	0.618–1.672	0.949
Unknown	1.228	0.296–5.086	0.777	4.436	0.492–39.950	0.184
Grade						
II	1	Reference		1	Reference	
III–IV	1.070	0.602–1.903	0.818	0.994	0.515–1.918	0.986
Unknown	1.510	0.732–3.111	0.264	1.172	0.520–2.643	0.701
Location						
Upper thoracic esophagus	1	Reference		1	Reference	
Middle thoracic esophagus	0.523	0.181–1.509	0.231	0.928	0.234–3.683	0.915
Lower thoracic esophagus	0.490	0.178–1.353	0.169	1.023	0.262–3.993	0.974
Unknown	0.556	0.156–1.985	0.366	0.920	0.176–4.805	0.921
T						
T1	1	Reference		1	Reference	
T2	0.997	0.492–2.020	0.993	1.793	0.798–4.029	0.158
T3	0.674	0.416–1.092	0.109	1.048	0.598–1.835	0.871
T4	1.488	0.817–2.708	0.193	1.474	0.716–3.034	0.292
N						
N0	1	Reference		1	Reference	
N1	3.560	1.073–11.810	0.038	1.119	0.206–6.079	0.896
M						
M0	1	Reference		1	Reference	
M1	2.106	1.374–3.228	0.001	1.372	0.776–2.425	0.277

## Data Availability

The data sets generated and analyzed were obtained in the SEER database in April 2022 (https://seer.cancer.gov/). These data sets are available from the corresponding author upon reasonable request.

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
