# Peer review of "Clinical-Pathological Characteristics of Adenosquamous Esophageal Carcinoma: A Propensity-Score-Matching Study"

_jpm, 2023, doi:10.3390/jpm13030468_

Round 1

Reviewer 1 Report

The EC study from a statistical point of view is a useful tool in the absence of an in-depth understanding of some complex phenomena. Among these, the authors mentioned tumor clonality or multiclonality.

In order for the authors to receive scientific recognition through citations, it would be useful for the authors to extrapolate the raw data (including the raw data resulting from the statistical analysis) into pathophysiological correlation data, so that the article can be useful to a wider audience.

For example, the authors mention the distribution mainly on the lower esophagus of EC, as well as the differentiated disposition in the thickness of the esophagus of different tumor clones, respectively the different predisposition to lymph node invasion and distant metastasis. From the point of view of cancer as a result between genetic and environmental factors, it can be assumed that the environmental factors in the lower esophagus are mainly of origin in the digestive tube (and less from atmospheric pollutants), of the stasis or reflux type . The link between gastro-oesophageal reflux and cancer is known, but it can be easily interpreted statistically if this link is also correlated with other factors, for example hiatal hernia which can decrease the vascularization in the depth of the thickness of the esophagus, with the explanation of the dependence on the tumor clone of the disposition tumors, beyond the easy explanation of the sources of cells that can be transformed into tumors.

Being a retrospective study, its conclusions should be able to modulate clinical conduct (diagnostic or therapeutic). In the absence of such evidence, it would be useful to extrapolate all statistical methods to highlight whether elements change over time that can lead to the modulation of therapeutic behavior compared to 10 years ago, for example. Partially the authors mentioned this, but they must identify at least one recommendation, even if apparently without major impact, such as avoiding food factors (identified statistically, even in other people's statistics, because it is not the objective of this article).

In short, the article must also present the data that are not statistically correlated but whose testing has been done (that is, at least each with each). If any other statistical correlation can be found apart from the ones presented, at least one hypothesis to explain them must be sought.

Author Response

Dear Reviewer 1,

Thanks for your timely and professional review.

This study retrospectively analyzed the clinical features and prognostic factors of esophageal adenossquamous carcinoma, and explored the survival of different treatment modes, hoping to guide clinical decision-making. According to your valuable suggestions, risk factors of different pathological types of esophageal cancer are added to the content of this article, and the possible pathogenesis of esophageal cancer is explored from the perspectives of environment, gender, diet, and location of the disease (The second paragraph of the discussion section). In addition, based on the results of the survival analysis, it is emphasized that surgery combined with chemoradiotherapy is preferred over other treatment modalities, and the potential risks of chemoradiotherapy are emphasized. The emerging tumor immunotherapy in recent years is briefly introduced (Line 351). Third, we analyzed the reasons for some non-statistically significant factors in the analysis results (Line 366). Finally, we emphasize the importance of cancer prevention and propose corresponding measures (Line 277).

Reviewer 2 Report

Dear author, I had the pleasure to review your study.
The subject is topical, the methodological choice adequate, the results clear and well presented, the reading pleasant and fluid.

On the other hand, methodologically, one point drew my attention:
the missing data and their management.
Among the matching variables, in particular, location and grade, after proportion score, you report for ADSC, SCC and ADC, for grade (G), respectively, a rate of 15.3%, 13.9% and 13.1% of missing data.

I think that multiple amputations are necessary and that the conclusions should be represented according to the results.
An additional chapter in the methodological session concerning the treatment of missing data could give additional power to your study.

Sincerely  

Author Response

Dear Reviewer 2,

Thanks for your constructive suggestions and comments.

Due to the inclusion criteria set, we lost a lot of patient data with incomplete information. Considering the possible impact on statistical results, we used the original data again for survival analysis (see supplementary chart). The overall results were basically consistent with those after the screening, which verified the reliability of the results.

According to your comments, we revised and improved the manuscript.

Thank you!